# New Genetic Variants of *RUNX2* in Mexican Families Cause Cleidocranial Dysplasia

**DOI:** 10.3390/biology13030173

**Published:** 2024-03-08

**Authors:** Jaime Toral López, Sandra Gómez Martinez, María del Refugio Rivera Vega, Edgar Hernández-Zamora, Sergio Cuevas Covarrubias, Belem Arely Ibarra Castrejón, Luz María González Huerta

**Affiliations:** 1Department of Medical Genetics, Centro Médico Ecatepec ISSEMYM, Ecatepec 55000, México State, Mexico; jaimetor77@gmail.com; 2Servicio de Genética, Hospital General de México “Eduardo Liceaga” (HGM), México City 06720, Mexicocuqui13rivera@gmail.com (M.d.R.R.V.); sergiocuevasunam@gmail.com (S.C.C.); ibarra.arely.c@gmail.com (B.A.I.C.); 3Medicina Genómica, Instituto Nacional de Rehabilitación “Luis Guillermo Ibarra Ibarra”, México City 14389, Mexico; edgarhz1969@yahoo.com.mx

**Keywords:** cleidocranial dysplasia, CCD, *RUNX2* gene, novel mutations, Runt-related transcription factor 2 gene

## Abstract

**Simple Summary:**

Cleidocranial dysplasia is a rare disease, manifested by anomalies in the skull, face, teeth, thorax, clavicle, extremities, and short stature. The disease is caused by mutations in the *RUNX2* gene, which is involved in the differentiation of cells that give rise and formation of bones. In this study, the genetic material of four patients and their families was analyzed, with the purpose of identifying changes in the sequence of the *RUNX2* gene, finding three new changes and one change already reported in the literature. One patient presented cataract and damage to the retina of the eye, and data were not reported in other patients. A bioinformatic analysis of the RUNX2 protein was carried out with the aim of predicting mechanisms, such as the function, stability, and conformation of the protein, and of trying to understand its relationship with the variable presentation of signs and symptoms in this condition. These results can be useful in the genetic counselling of patients.

**Abstract:**

Cleidocranial dysplasia (CCD) is an autosomal dominant skeletal dysplasia characterized by persistent open skull sutures with bulging calvaria, hypoplasia, or aplasia of clavicles permitting abnormal opposition of the shoulders; wide public symphysis; short middle phalanx of the fifth fingers; and vertebral, craniofacial, and dental anomalies. It is a rare disease, with a prevalence of 1–9/1,000,000, high penetrance, and variable expression. The gene responsible for CCD is the Runt-related transcription factor 2 (*RUNX2*) gene. We characterize the clinical, genetic, and bioinformatic results of four CCD cases: two cases within Mexican families with six affected members, nine asymptomatic individuals, and two sporadic cases with CCD, with one hundred healthy controls. Genomic DNA analyses of the *RUNX2* gene were performed for Sanger sequencing. Bioinformatics tools were used to predict the function, stability, and structural changes of the mutated RUNX2 proteins. Three novel heterozygous mutations (c.651_652delTA; c.538_539delinsCA; c.662T>A) and a previously reported mutation (c.674G>A) were detected. In silico analysis showed that all mutations had functional, stability-related, and structural alterations in the RUNX2 protein. Our results show novel mutations that enrich the pool of *RUNX2* gene mutations with CCD. Moreover, the proband 1 presented clinical data not previously reported that could represent an expanded phenotype of severe expression.

## 1. Introduction

Cleidocranial dysplasia (CCD; OMIM 119600) is a rare disease (ORPHA 1452), presenting with short stature, excessive mobility of the shoulders, impaired bone growth, skeletal and craniofacial anomalies, hypertelorism, a general midface retrusion, and a mandible prognathism. It is associated with teeth abnormalities, such as supernumerary teeth, failure of eruption, and multiple dental anomalies. It is an autosomal dominant syndrome, with a prevalence of 1–9/1,000,000, high penetrance, and variable expression [1,2].

Cleidocranial dysplasia is caused by a heterozygous loss-of-function mutation of the Runt-related transcription factor 2 gene (*RUNX2;* OMIM 600211). The *RUNX2* gene located on chromosome 6p21 encompasses a region of 223 kb with eight exons and encodes a polypeptide of 521 amino acids (NP_001019801.3) [3]. This polypeptide comprises the first activation domain (1) [3], the polyglutamine/alanine (QA) domain, Runt homologous domain (RHD), nuclear localization signal (NLS) region, proline/serine/threonine (PST)-rich region, nuclear matrix-targeting sequence (NMTS) domain [4], and the conserved repressive motif (CRM), consisting of the amino acids valine-tryptophan-arginine-proline-tyrosine (VWRPY) [5]. *RUNX2* has two promoters that result in the expression of two isoforms: the distal promoter (P1) generates the canonical isoform 1 or type II RUNX2 mRNA (NM_001024630.4) and the proximal promoter (P2) generates type I RUNX2 mRNA [6]. The RUNX2 protein is a transcription factor and essential for osteoblast differentiation during intramembranous and endochondral ossification [7,8,9].

CCD can be diagnosed with clinical and radiological evaluation and validated by molecular studies. Heterozygous loss of function *RUNX2* gene, which plays an important role in osteogenesis and differentiation of precursor cells, causes a CCD phenotype [10]. More than 230 *RUNX2* gene mutations have been associated with CCD [11,12]. In a present study, we characterized the clinical, genetic, and bioinformatic results of three novel and one previously reported mutation in the *RUNX2* gene in two familial and two sporadic cases with CCD and described clinical data not previously reported.

## 2. Material and Methods

### 2.1. Cases Studied

Two cases within Mexican families (Family 1 and 4) with six affected members, nine asymptomatic individuals, and two sporadic cases (Family 2 and 3), in addition to one hundred healthy controls, were included in this study. The probands were three males and one female. Clinical and all other relevant data of the probands are shown in Table 1. A severe phenotype was defined as the presence of three major features (cranial, clavicular, and dental anomalies) and the presence of any minor feature (short stature, facial, thoracolumbar, extremity, or other anomalies). A moderate phenotype was defined as two major features with any minor feature, and a mild phenotype was defined as one major feature with any minor feature. As there is no severity classification for CCD in the literature, this is the authors’ proposal based on the frequency of clinical alterations in cases reported in the literature. Proband 1 (Figure 1A–F) and his mother showed a similar phenotype except that mother was absent of cataract, retinal damage, or facial changes. Data on the aunt were not obtained; it was only known that she had a diagnosis of CCD. With respect to proband 4, her brother and mother had similar characteristics without any headache, brachydactyly of fourth phalanx of the left foot, or facial anomalies. The parents of probands 2 and 3 showed no clinical abnormalities. All individuals agreed to participate in this study and provided informed consent.

### 2.2. Direct Sequencing of the RUNX2 Gene

After genomic DNA extraction using conventional methods, the *RUNX2* gene was amplified via a polymerase chain reaction. We designed primers flanking all 8 coding exons and intron-exon boundaries of the *RUNX2* gene using Primer3 (http://bioinfo.ut.ee/primer3-0.4.0/) URL (accessed on 17 May 2023). Primer sequences and PCR conditions are shown in Table 2. PCR products obtained from all affected and non-affected members of the families and 100 normal controls were analyzed through Sanger sequencing on an ABI 377 automated sequencer (PE Biosystems, Foster City, CA, USA). All DNA analyses were performed twice.

### 2.3. In Silico Prediction Analysis

The comparative analysis of the evolutive conservation was carried out with phyloP and phastCons. To assess the functional affection of the identified heterozygous variants, Polymorphism Phenotyping v2 (PolyPhen 2.0), MutationTaster 2021 (www.mutationtaster.org (accessed on 17 May 2023)), PROVEAN 1.1.5 (https://provean.jcvi.org/index.php (accessed on 17 May 2023)), and SIFT tools were used. To predict protein stability changes, GROMOS96-Swiss-Pdb Viewer, and I-Mutant 2.0 (https://folding.uib.es/cgi-bin/i-mutan t2.0.cgi (accessed on 17 May 2023)) were used. The change in the force field energy was represented as the logarithm of the energy (Log). RUNX2 3D protein prediction was determined via the alignment of the protein from the RaptorX server [15], and the secondary structure’s changes and homology modeling were observed in Swiss-Pdb Viewer, version 4.0, using SWISS-MODEL [16].

### 2.4. Ethical Aspects

The individuals with CCD were selected under the official Mexican standard NOM-253-SSA1-2012 guidelines [17], and all participants received oral and written information about this study. The study protocol was reviewed and approved by the Ethics and Research Committee of the Centro Medico ISSEMYM Ecatepec (approval number [PICME92]) and Hospital General de Mexico (approval number [DI/23/501/04/32]).

## 3. Results

Analysis of the *RUNX2* gene evidenced three novel mutations and one that was previously reported. The proband and the mother of Family 1 had the c.651_652delTA mutation in exon 4 (Figure 2A); this mutation changes the amino acid lysine to serine at Position 218 and creates a premature termination 17 codons downstream of the deletion site (p.Lys218Serfs*17). The proband of Family 2 presented the c.538_539delinsCA mutation in exon 3 that generates the substitution of glutamine instead of alanine (p.Ala180Gln) (Figure 2B). The proband of Family 3 harbored the recurrent missense mutation in exon 4 c.674G>A that results in the change of glutamine via arginine, p.Arg225Gln (Figure 2C). The parents of probands 2 and 3 did not show the new variants. The proband, brother, and mother of Family 4 showed the missense mutation c.662T>A in exon 4 that produces the substitution of glutamate via valine, p.Val221Glu (Figure 2D). Relevant genetic and protein data are shown in Table 1. These mutations were discarded in the *RUNX2* gene in the unaffected members of the family and 100 healthy controls. No other relevant nucleotide variations or polymorphisms were detected in the rest of the analyzed exons. The novel variants were not found in the genomAD, 1000 Genomes, or MutationTaster databases.

The comparative analysis using phyloP and phastCons of the respective nucleotide’s evolutive conservation in the human *RUNX2* gene compared with that of one other species had a score of 4.5 to 6.0 and 1 for the c.651_652delTA, c.674G>A, c.538_539delinsCA, and c.662T>A variants, respectively. This suggests that these amino acids are highly conserved, indicating a deleterious effect of these changes on the RUNX2 protein. The analysis prediction of deleterious effects in protein function is shown Table 3.

An illustrative image is included locating the positions of the variants found in our patients within exons 3 and 4 of the *RUNX2* gene, as well as the position of the amino acid changes within the RHD domain of RUNX2; in addition, an image shows the structure secondary with their domains, respectively (Figure 3).

The total energy of each of the three mutants of RUNX2 protein was high, with log 18.45, 15.00, and 15.40 KJ/mol, respectively; it was the highest compared to the wild-type (log 14.48 KJ/mol). The deletion p.Lys218Serfs*17 showed a low energy value of 12.51 KJ/mol. These high and low energy levels implied an underlying damaging effect on protein structure, thereby affecting protein stability and function. Free-energy change (DDG) analysis with I-Mutant 2.0 showed negative values from −1.03 to −1.72 DDG in the three novel missense variants. This denotes decreasing protein stability with a deleterious effect (Table 3).

Structural change predictions of the mutant forms of RUNX2 protein showed that the amino acid changes produced loss and gain of hydrogen bonds, of α-helix regions, and of β-leaves (Table 3), leading to protein conformational modifications (Figure 4A–C and Figure 5A–D).

## 4. Discussion

It is important to note that CCD has variable clinical expressiveness, and particular clinical characteristics have been described in various articles, with the following frequency of manifestations: 93% cranial, 50% facial, 89% dental, 92% clavicular, 40% chest, 40% extremities, and 73% other anomalies (such as reduced height) [5,15,16,18,19,20,21,22]. In this study, two sporadic and two familial cases with CCD were analyzed. Sporadic cases represent around 40–60% of CCD cases [13,19]. Particularly, this has directed our attention to describing the proband of Family 1 because it presents different phenotypic characteristics than those commonly described. It is a 12-year-old male heterozygous for c.651_652delTA in exon 4 (Figure 1A). He exhibited left cataract with detachment of the retina, low hair implantation, bushy eyebrows and eyelashes, strabismus, eyelids turned down, and a wide mouth (Table 1). These clinical characteristics had not been previously described in individuals with CCD, which suggests makes us think that it could be a more severe case, and this could represent an expansion of expression of the *RUNX2* clinical phenotype–genotype association in CCD.

*RUNX2* gene mutations can be identified in 71% of CCD cases, with 60% of them being point mutations [23]. Around 50 missense mutations have been identified in the Runt domain [23]. Similarly, the mutations observed in this study’s probands were found inside the Runt domain of the RUNX2 protein; three in exon 4 and one in exon 3 (p.Ala180Gln) of the *RUNX2* gene. The recurrent point mutation p.Arg225Gln, present in proband 3, was previously found to abolish the function of the NLS with accumulation of RUNX2 protein in the cytoplasm [15,18], while the p.Arg225Trp and the p.Arg186Thr mutations (located close to the p.Ala180Gln site of our Case 2), previously reported, diminish the DNA-binding ability [24]. In contrast, previous reports indicate that the p.Arg186Thr and p.Leu113Arg mutations alter the transactivation activity of osteocalcin, collagen I, bone sialoprotein, and osteopontin genes [22], whereas the p.Leu62 of mouse RUNX1 homologous to p.Leu113 of human RUNX2 protein is involved in altered stabilization producing a defective DNA binding and heterodimerization with core binding factor β (CBFβ) [24], also called its partner subunit or the polyomavirus enhancer-binding protein 2b (PEBP2b) [21]. Proband 4 presented a p.Val221Glu change located on the same site of a previous case who presented a p.Val221Gly mutation. A familial case with moderate CCD [25], proband 4, showed a severe phenotype, while her brother and her mother were less severely affected. We consider that in the p.Ala180Gln and p.Val221Glu mutations, an alteration was caused due to the accumulation of RUNX2 protein in the cytoplasm or an alteration in DNA binding due to its proximity to p.Arg186Thr, p.Arg225Gln, and p.Arg225Trp [15,18,24].

Proband 1 with the c.651_652delTA (p.Lys218Serfs*17) mutation had a family history of CCD and a severe phenotype. His mother and his aunt had a less severe phenotype. The c.651_652delTA mutation was located near a previously reported small deletion CCTAdel635_638 in exon 3 of *RUNX2*, which led to a stop codon in the 220 amino acid. This frameshift mutation caused the deletion from the NLS, possibly affecting the accumulation of the RUNX2 protein in the nucleus, with the entire family presenting typical CCD [20]. A 2-year-old boy with classical CCD showed a heterozygous deletion (c.593_601delCCTTGACCA, p.Thr198_Thr200del) in the Runt domain; the 3D modeling assessment demonstrated that this mutation abolished heterodimerization of the RUNX2 protein with its partner subunit: the polyomavirus enhancer-binding protein 2b (PEBP2b); in this report, the transactivation study showed that the p.Thr198_Thr200del and p.Leu199Phe mutants had significantly lower transcription activities (44% and 63% of the wild type, respectively) [21]. A previous report of a frameshift deletion was observed in a 15-year-old boy who presented a c.1550delT (p.Trp518Glyfs*60) mutation [26]. Another case of a 10-year-old boy with his mother showed a frameshift deletion with a c.1554delG (p.Trp518Cysfs*61) mutation [27], and the authors in both studies concluded that these mutations affected the transactivation activity in the C-terminal zone of RUNX2. All individuals in these studies presented typical CCD features. The mutation in proband 1 led to a stop codon in the 235 amino acid sequence at the end of the NLS region, modifying the Runt and the NLS sequences and losing the PST region and the NMTS domain. This indicates that the effect produced by these mutations could be of the negative dominance type. Other studies on the p.Gln67* (QA domain) and p.Gly462* (C-terminal PST region) mutations showed that the subcellular distribution and the transactivation activity on its downstream target genes were affected, thus lowering protein stability and affecting the expression of bone marker genes and osteoblast differentiation in CCD cases [5,22]. The mutation p.Lys218Serfs*17 of proband 1 would similarly affect its nuclear distribution, binding to DNA, or transactivation of target genes.

No genotype–phenotype correlation is present in CCD due to the large variable clinical expression. For example, in the family with the report of the mutation p.Thr200Ala, the father just showed dental abnormalities, whereas his two children exhibited classic CCD. In the same study, an electrophoretic mobility shift assay revealed that p.Thr200Ala retained almost normal transcription activity with no affectation on binding to DNA or transactivation activity; possibly, the mutation produced a neomorphic effect, altering different functions of the Runt domain [15]. Another study exhibited a group of cases with an unaffected Runt domain that presented features of CCD and a milder short stature compared to an affected Runt domain group with partial CCD phenotype but significantly short stature [16]. In contrast, a study found that short stature was more evident in the unaffected Runt domain, which was predicted to interfere with the transactivation activity of its target genes [19]. In our study, two cases had short stature and other two cases had normal stature, all involving the Runt domain.

In our study, the in silico analysis predicted that all mutations affect the RUNX2 protein function. In addition, the force field energy of the mutant RUNX2 proteins showed significant deviations of the energy per amino acid change compared to the RUNX2 wild-type. Molecular dynamics analysis with I-mutants showed negative values; the changes with high and low energies implied a damaging effect on protein stability and function. Thus, this bioinformatic analysis is in agreement with previous functional studies where changes in amino acids and the truncation mutation were involved in impaired protein stabilization and subsequent defect in DNA-binding ability, altering heterodimerization with core binding factor β (CBFβ) [24]. The union of RUNX2 with its partner; cofactor CBFβ, is very important since it allows to improve its DNA binding affinity by promoting structural changes that reveal the RUNX2-DNA’s interaction surface [28], it also protects and stabilizes against proteolytic degradation [29]. Some factors, such as Msh homeobox 2 (Msx2), that twist or promyelocytic leukemia zinc finger protein (PLZF) and osterix (Osx) and participate in the induction of RUNX2 expression [30,31] can be affected by their poor interaction with the Runt domain. Likewise, RHD damage downregulates transactivation activity, decreasing the expression of bone matrix genes, including collagen I, osteocalcin, bone sialoprotein, and osteopontin [3,5,24].

Finally, the generation of 3D structures of the wild-type and mutant proteins revealed significant changes in the secondary structures with a gain or loss of hydrogen bonds, the α-helix regions, and the β-leaves. These changes showed a folded protein, suggesting a modification of the protein–protein interactions, as previously observed [21]. The above data suggest that variations in protein accumulation in the cytoplasm, DNA binding, transactivation activity of RUNX2 at its target genes, or heterodimerization with CBFβ lead to altered transcription activities and variable degrees of haploinsufficiency or protein activity [19], having an important role in the variable clinical expression of CCD [15]. In addition to the internal factors in the previously mentioned protein, we can consider that the variability in the phenotype can also be influenced by stochastic factors, specific to each individual, such as the combination of single nucleotide polymorphic variants, epigenetics, modifier genes, copy number variations, and their interaction with the environment [32], which could also lead to the variable clinical expression of diseases with an autosomal dominant inheritance pattern.

In conclusion, the three novel mutations identified in these cases within Mexican families with CCD enriched the pool of *RUNX2* gene mutations. This study supports previously published data, suggesting that the alteration of stability and the conformational change of RUNX2 protein play a significant role in the interaction with other proteins, affecting its transcription or various functions and thus the variable expression of CCD [25]. Moreover, additional clinical features were observed in one case, such as cataract with detachment of the retina, low hair implantation, strabismus, eyelids turned down, and a wide mouth, all of which could represent an expansion of clinical phenotype–genotype’s expression in cleidocranial dysplasia.

## Figures and Tables

**Figure 1 biology-13-00173-f001:**
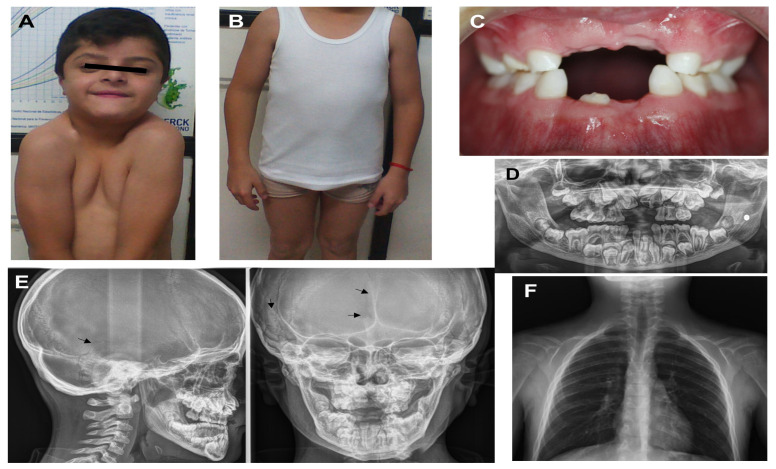
Appearance of proband 1 at the age of 12 years: (**A**) facial features present, such as bushy eyelashes and eyebrows, strabismus, hypertelorism, eyelids turned down, anteverted nostrils, long philtrum, wide mouth, and easy positioning of the shoulders toward the center; (**B**) cubitus valgus and genu varum; (**C**) hypodontia and oligodontia; (**D**) X-ray panoramic view shows supernumerary teeth and delayed eruption of permanent teeth; (**E**) X-ray cranial lateral and frontal show mild open suture at the temporal, frontal region (black arrow), and Wormian bones at the temporal region (black arrow); (**F**) clavicle hypoplastic.

**Figure 2 biology-13-00173-f002:**
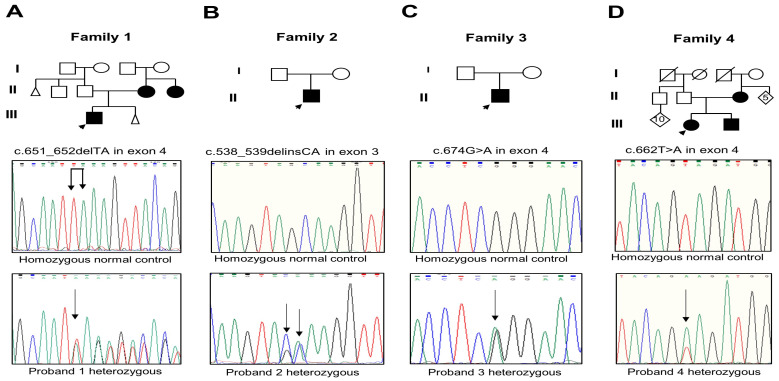
(**A**–**D**) Pedigree of Families 1 to 4 with the partial electropherogram of the healthy control and of the c.651_652delTA (p.Lys218Serfs*17), c.538_539delinsCA (p.Ala180Gln), c.674G>A (p.Arg225Gln), and c.662T>A (p.Val221Glu) mutations, respectively. Black arrow indicates nucleotide changes of sequences in probands compared to healthy controls.

**Figure 3 biology-13-00173-f003:**
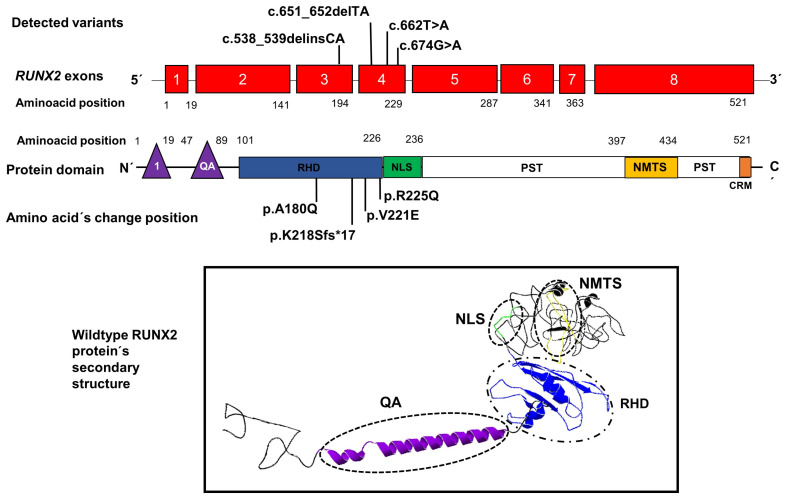
Position of novel genetic variants in the exons 3 and 4 (red boxes) of the *RUNX2* gene and the position of amino acid changes in the domain RHD (blue rectangle) of the RUNX2 protein. The lower image shows the secondary structure of the protein with its domains QA: glutamine/alanine domain (purple), RHD: Runt homologous domain (blue), NLS: nuclear localization signal region (green), and NMTS: nuclear matrix-targeting sequence domain (yellow). 1: the first activation domain (purple triangle), CRM: the conserved repressive motif (orange rectangle).

**Figure 4 biology-13-00173-f004:**
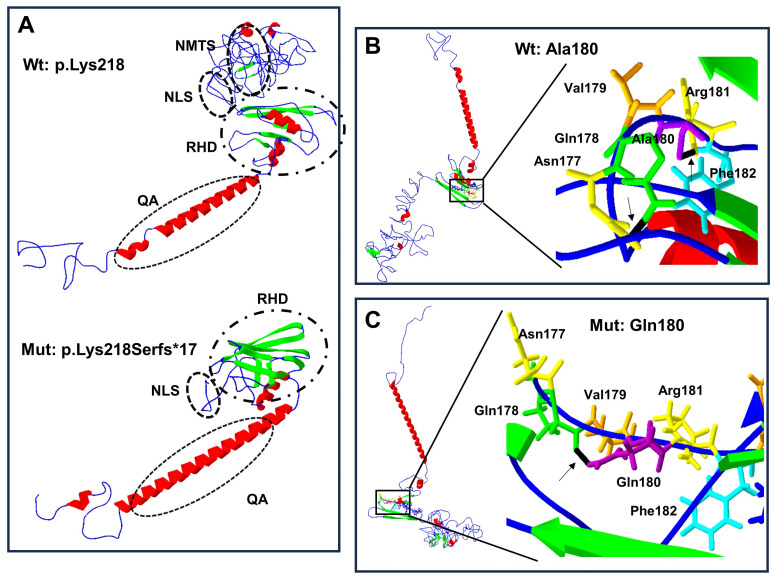
Protein structure prediction of p.Lys218Serfs*17 and p.Ala180Gln-mutated human RUNX2. The α-helix (red), β-leaf (green), coils, and loops (blue) are shown. (**A**) The p.Lys218Serfs*17 mutation (below) causes modification of RHD and NLS with loss of PST, NMTS (inside the ellipses), and CRM is present in the wild-type (Wt) p.Lys218 protein (above). (**B**) Wild-type Ala180 protein (left side) adopts a different folded conformation in the (**C**) p.Gln180 mutated protein (left side). (**B**) Enlarged image: the wild-type p.Ala180 (purple) has a bridge of hydrogen (black line marked with arrow) link with Phe182 (light blue), while in the mutated (**C**) enlarged image, p.Gln180 (purple) has a hydrogen bond (black line marked with arrow) attached with p.Gln178 (green).

**Figure 5 biology-13-00173-f005:**
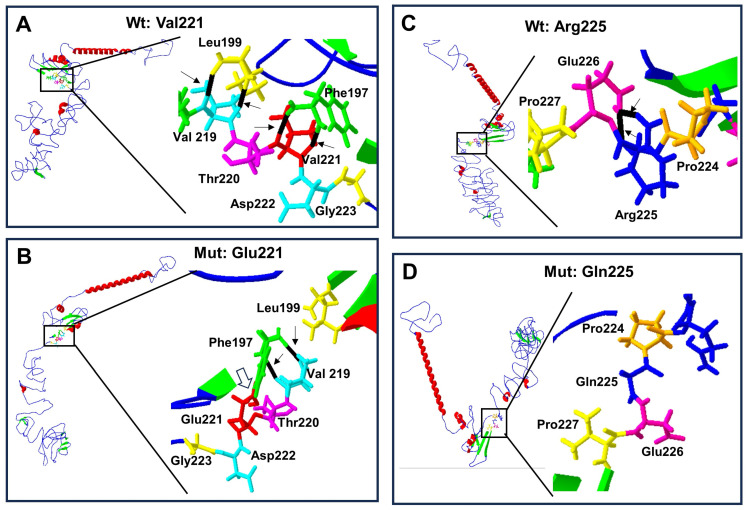
Protein structure prediction of p.Val221Glu and p.Arg225Gln-mutated human RUNX2. Wild-type protein ((**A**) and (**C**) top left) acquires a folded conformation in the mutated proteins (**B**) and (**D**) down left), respectively. The α-helix (red), β-leaf (green), and coils/loops (blue) are shown. Enlarged image shows that (**A**) the p.Val221 (red) has two bridges of hydrogen linked with p.Phe197 (green), while (**B**) p.Glu221 (red) loses the two hydrogen bonds with p.Phe197 but acquires a covalent bond between them (open arrow); additionally, p.Phe197 links two hydrogens bonds with p.Val219 (light blue). Enlarged image: (**C**) the p.Arg225 (blue) has two hydrogen bonds linked with p.Gln226 (pink), while (**D**) p.Gln225 (blue) loses its two hydrogen bonds.

**Table 1 biology-13-00173-t001:** Relevant clinical and *RUNX2* gene mutations data of the four probands with cleidocranial dysplasia.

Features	Case 1	Case 2	Case 3	Case 4
Age (y)	12	1	7	29
Sex	M	M	M	F
Cranial abnormalities	Low hair implantation; Wormian bones; mild open fontanelle/sutures.	Prominence frontal and parietal; broad fontanelle and sutures.	Frontal prominent; broad sutures and fontanelles.	Brachycephaly; broad forehead with medium cleft; cranial bone thickening; Wormian bones; headache.
Facial abnormalities	Populated eyelashes/eyebrows; strabismus; hypertelorism; eyelids turned down; anteverted nostrils; long philtrum; wide mouth.	Hypertelorism.	Midface hypoplasia; hypertelorism; wide/low nasal bridge; bulbous nasal tip; long philtrum.	Midface asymmetry and hypoplasia; nasal deviation; hypertelorism.
Oral-dental abnormalities	Oligodontia and supernumerary teeth.	Central and lateral teeth present apparently normal.	Yellow and misaligned teeth.	Molars absent; supernumerary teeth.
Clavicle abnormalities	Hypoplasia.	Agenesis.	Agenesis.	Hypoplasia.
Thorax and spine abnormalities	Pectum excavatum and thoraco-lumbar scoliosis.	Absent.	Pectum excavatum and thoraco-lumbar scoliosis.	Short and wide chest; thoracic scoliosis.
Limbs abnormalities	Absent.	Absent.	Absent.	Brachydactyly 4th phalanx left foot. Cubitus valgus; genu varum.
Others abnormalities	Cataract, detachment of retina.	Absent.	Short stature; short neck.	Short stature; short neck with limited rotation.
Phenotype expression	Severe.	Moderate.	Severe.	Severe.
Family history	(+)	(−)	(−)	(+)
DNA change	c.651_652delTA;	c.538,539delinsCA	c.674G>A	c.662T>A
Exon location	4	3	4	4
Protein change	p.Lys218Serfs*17	p.Ala180Gln	p.Arg225Gln	p.Val221Glu
Domain location	RHD/NLS.	RHD	RHD	RHD
Reference	Novel.	Novel.	[13,14]	Novel.

Y: years, M: male, F: female, absent: phenotype absent, RHD: Runt homologous domain, NLS: nuclear localization signal.

**Table 2 biology-13-00173-t002:** Primers to amplify the *RUNX2* gene.

Exon Number	Primers	Size	Tm
2	F 5′-GGCCACTTCGCTAACTTGTG-3′ R 5′-GTAGCCTCTTACCTTGAAGG-3′	422 pb	56 °C
3	F 5′-GGACTAGAACACTAAGTCCTG-3′ R 5′-CACTCAACTTCATCTGGATG-3′	419 pb	60 °C
4	F 5′-CATTGCCTCCTTAGAGATGC-3′ R 5′-ATTCCTCATAGGGTCTCTGG-3′	280 pb	60 °C
5	F 5′-GCAT GGTCAATTGTTCAGCT-3′ R 5′-CTGCCAGCGTCTATGCAAG-3′	273 pb	60 °C
6	F 5′-GGCTGCAATGGTTGCTATAC-3′ R 5′-TGTGAGCATGGATGAGACAG-3′	289 pb	60 °C
7	F 5′-CATAGAACATTAGAGCTGGAAGG-3′ R 5′-CTCACAAAATCGGACAGTAAC-3′	199 pb	60 °C
8	F 5′-GGTGCATTTGAAGGTCTGTC-3′ R 5′-ATTGATACGTGTGGGATGTGG-3′	677 pb	60 °C

**Table 3 biology-13-00173-t003:** Results of in silico analysis of wild-type and mutated RUNX2 proteins.

Protein/Amino Acid ID	PolyPhen2/MutationTaster/PROVEAN/SIFT (Score)	Log of Force Field Energy (KJ/mol)	I-Mutant 2.0	Secondary Structure Changes			
				Structural Molding	# of α-Helix	# of β-Leaf	# of Hydrogen Bonds
RUNX2 wild-type	-	14.48	-	-	6	7	NA
RUNX2 p.Gln180	NA	18.45	NA	Folded	5	11	NA
RUNX2 p.Glu221	NA	15.00	NA	Folded	4	11	NA
RUNX2 p.Gln225	NA	15.40	NA	Folded	7	9	NA
p.Lys218Serfs*17	1.000/disease causing (1.0)/NA/NA	12.51	NA	Truncated	3	8	NA
Ala180	-	5.53	-	NA	NA	NA	0
Gln180	1.000/disease causing (0.99)/deleterious (−4.66)/damaging (0.00)	6.24	−1.72	NA	NA	NA	1
Val221	-	2.59	-	NA	NA	NA	2
Glu221	1.000/disease causing (0.99)/deleterious (−5.85)/damaging (0.00)	7.99	−1.69	NA	NA	NA	0, Gain of a covalent bond
Arg225	-	7.99	-	NA	NA	NA	2
Gln225	0.980/disease causing (0.99)/deleterious (−3.8)/damaging (0.03)	6.46	−1.03	NA	NA	NA	0

NA: Not applicable.

## Data Availability

All relevant data used in this study were included in this manuscript. The corresponding author can be contacted if any further information is needed. The datasets generated and/or analyzed during this study are available from the corresponding author upon reasonable request.

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
