# Peer review of "New Genetic Variants of RUNX2 in Mexican Families Cause Cleidocranial Dysplasia"

_biology, 2024, doi:10.3390/biology13030173_

Round 1
Reviewer 1 Report
Comments and Suggestions for Authors
Shown in the file.

Comments on the Quality of English LanguageShown in the file.
Author Response
Thank you very much for taking the time to review this manuscript. Please find the detailed responses below and the corresponding revisions/corrections highlighted/in track changes in the re-submitted files.
Reviewer 1
The authors presented cleidocranial dysplasia (CCD), an autosomal dominant skeletal disorder characterized by various skeletal anomalies. The runt-related transcription factor 2 (RUNX2) gene is responsible for CCD. Genomic DNA analysis revealed three novel mutations and one previously reported mutation in the RUNX2 gene. The results add new mutations to the pool of CCD-associated mutations in the RUNX2 gene and suggest an expanded phenotype in a patient with previously unreported clinical data.
Bioinformatics tools were used to predict the functional, stability, and structural changes in the mutant RUNX2 proteins, which were altered in all cases.
Overall the reviewer thinks the evidence provided by the authors is convincing in the mutant RUNX2 proteins, although some issues should be addressed to clarify the results. The reviewer also regrets that there is a lack of attention to the content of the paper writing.
Reviewer 1
- Results (Figures 2 and 3): It is difficult to see the exact differences between wild-type and mutants because the authors use different orientations and magnifications of the figure panels. The authors must revise the Figures 2 and 3 to reflect the author's explanations.
1) When comparing wild-type and mutant protein structures, the differences should be evident in the same location, at the same magnification, and in the same view from the same orientation.
Now we have eliminated previous ones (Figures 2 and 3) and we change new images (Figure 4 and 5) standardizing the same location, magnification, view from the same orientation, however, some had to be rotated for a better appreciation of some images such as hydrogen bonds.
2) The authors use the wrong panels in Figure 2A since a low magnification panel appears to be wild-type, but a high magnification panel labels the mutant.
We have reviewed Figure 2 (Now Figure 4) confirming that the low magnification panel is wildtype or mutant and the high magnification panel corresponds to the expansion of the protein site, where the wildtype or mutant amino acid change is located.
3) Please use arrows etc. in the figures to clarify the intent of the author's explanations.
In the figures, we have added arrows to indicate hydrogen bonds or covalent.
4) It is difficult to see the blue amino acids because of the blue background. The authors should
change one of the colors.
Now we have changed the background to white for better color distinction.
5) It is difficult to distinguish between hydrogen bonds and covalent bonds. The colors and/or lines of bonds within figures must be defined in the figure legend.
In the figure legend the word “(arrow)” is added and in the figures the hydrogen bridges are highlighted more in black and are indicated with arrows.
- In silico analyses can predict the functional effects of mutant proteins, but functional assays
remain essential.
We appreciate your valuable comment and suggestion, which is considered for future studies.
- It would be beneficial for the reader to include a schematic diagram illustrating the structure of the RUNX2 proteins with amino acid numbers highlighting the specific mutations being discussed. This visual aid may enhance understanding and provide a clear reference point for the information presented.
Where is NLS located in the RUNX2 proteins?
Now we have added a schematic diagram (Figure 3) with the legend: “Figure 3. Position of novel genetic variants in the exons 3 y 4 (red boxes) of the RUNX2 gene and the position of amino acid changes in the domains RHD of the RUNX2 protein. The lower image shows the secondary structure of the protein with its domains QA (purple), RHD (blue), NLS (green), and NMTS (yellow).”
- Please revise Table 2 to transfer the accurate information. Case 1 is a 1-year-old boy. Therefore, (-), indicating a negative phenotype, is inappropriate for Dental abnormalities. Case 2 has thoraco-lumbar scoliosis, but is shown as a negative phenotype in Thorax Anomalies. “Pectus excavatum” does not belong to a limb. In addition, please include oral phenotypes such as cleft palate as traits for all cases.
Clinical data in Table 2 were now reviewed and corrected, the oral phenotype was included as part of the oral-dental anomalies, however, our patients did not show such a characteristic. The signs (+) or (-) were changed to a more appropriate term “Absent” in the table 2 and “Absent: phenotype absent” at the foot of the table 2.
Minor:
- Table 3: It is not possible to determine to which element each value or analysis corresponds because there are no ruled lines. What is [2, newline, 0], of # of hydrogen bonds of Val221?
In table 3, ruled lines have been added for greater clarity of the values of each element.
- Please ensure that gene names used in your manuscript have been approved by the HUGO
nomenclature committee. Ex) Italic RUNX2 indicates the human gene.
Now we have paid greater attention to writing the name of the RUNX2 gene in italic and the protein in non-italic, which is approved by the HUGO nomenclature committee.
- Table 2: There are extra spaces in the primer sequences or missing several bases.
The sequence of the forward oligonucleotide of exon 8 has now been corrected
- Is p.Lys218Sfs*17 the same as p.Lys218Serfs*17?
We have now corrected the variant name “p.Lys218Sfs*17” to “p.Lys218Serfs*17”
- Line 13: What do you mean by “whit”?
“whit” has been corrected to “with”
- Line 19: Replace “RUNX2 gene was” with “RUNX2 gene were”.
“RUNX2 gene was” has been corrected to “RUNX2 gene were”
- Line 26: Replace “sever” with “severe”. “sever” has been replaced by “severe”
- Line 141: Correct “a covalent bonds”. Corrected to “a covalent bond”.
- Line 160: Please cite the reference for “Around 50 missense mutations have been identified in the Runt domain.
Now reference “22” was added.
Reviewer 2 Report
Comments and Suggestions for Authors
In their work the authors provide an in silico analysis of mutations identified in Mexican CCD families. We have some suggestions to help the authors improve their manuscript:
1) please provide the declaration of approval of the study by an Ethics Committee
2) add a section to describe the clinical presentation of patients: indeed, notwithstanding Table 1, it is not clear why some patients are deemed more severe. Please consider also the possibility to add clinical pictures of the affected members of families where there is different severity of presentation. Which may be the reason for the variable severity? Can the authors speculate (and maybe also test any hypothesis, such as the presence of genetic modifiers)?
3) Table 1: in the description of clinical features, use the adjective "Absent" instead of "(-)". Regarding clavicle abnormalities, use the full description "hypoplasia" and "agenesis" instead of H and A.
4) Where the authors describe the results of mutation analysis with prediction software, please avoid verbs such as "reveal" (lines 124, 226) in order to make it clear to the readers that in fact it's a prediction and there is no formal demonstration.
5) please verify that the protein name is not in italics.
6) please rephrase the legend to Fig. 2 and to Fig. 3:
"(A) The wild-type (Wt) p.Lys218 protein loses the nuclear localization site, proline-serine-threonine-rich domain, nuclear matrix targeting signal, and the conserved repression motif (white) in the p.Lys218Sfs*17 mutated (Mut) protein." Instead we suggest "The p.Lys218Sfs*17 mutation causes loss of the nuclear localization site, proline-serine-threonine-rich domain, nuclear matrix targeting signal,..."
Similarly, for "(B) Wild-type Ala180 protein adopts a folded conformation in the (C) p.Gln180 mutated protein. The wildtype p.Ala180 (blue) without bridges of hydrogens, in the mutated p.Gln180 (blue) gain a bridge of hydrogen bonds with p.Gln178 (yellow)." we think it's more appropriate "the p.Gln180 mutation causes the acquisition of a folding conformation owing to the possibility of hydrogen bonds between p.Gln180 and p.Gln178"
Accordingly, modify also the legend to Figure 3.
Comments on the Quality of English Language
Please extensively and carefully revise English (in the abstract, there are several sentences lacking a verb, just to mention an issue as an example).
Author Response
Thank you very much for taking the time to review this manuscript. Please find the detailed responses below and the corresponding revisions/corrections highlighted/in track changes in the re-submitted files.
Reviewer 2
In their work the authors provide an in silico analysis of mutations identified in Mexican CCD families. We have some suggestions to help the authors improve their manuscript:
- please provide the declaration of approval of the study by an Ethics Committee ?
Declaration of approval of the study by an Ethics Committee were added
- add a section to describe the clinical presentation of patients: indeed, not with standing Table 1, it is not clear why some patients are deemed more severe.
Now in the patient’s section we add the text “A severe phenotype was defined as the presence of three major features (cranial, clavicular, and dental anomalies) plus the presence of any minor feature (short stature, facial, thoracolumbar, extremity, or other anomalies). A moderate phenotype was defined as two major features with any minor feature, and a mild phenotype was defined as one major feature with any minor feature. As there is no severity classification for CCD in the literature, this is the authors’ proposal based on the frequency of clinical alterations in cases reported in the literature.”
Please consider also the possibility to add clinical pictures of the affected members of families where there is different severity of presentation.
In the patient’s section we add a text summarizing the clinical data of the relatives of the probands carrying the variants (family 1 to 4): “The proband 1 (Figure 1A–F) and his mother showed a similar phenotype except that mother was absent of cataract, retinal damage, or facial changes. Data on the aunt were not obtained; it was only known that she had a diagnosis of CCD. With respect to proband 4, her brother and mother had similar characteristics without any headache, brachydactyly of fourth phalanx of the left foot, or facial anomalies. The parents of probands 2 and 3 showed no clinical abnormalities.”
Which may be the reason for the variable severity? Can the authors speculate (and maybe also test any hypothesis, such as the presence of genetic modifiers)?......
In the discussion section (in the end) added the text “In addition to the internal factors in the previously mentioned protein, we can consider that the variability in the phenotype can also be influenced by stochastic factors, specific to each individual, such as the combination of single nucleotide polymorphic variants, epigenetics, modifier genes, copy number variations and their interaction with the environment [27], which could also lead to the variable clinical expression of diseases with an autosomal dominant inheritance pattern”
Suda, N et al, 2010 was added to the reference list with number 27
Table 1: in the description of clinical features, use the adjective "Absent" instead of "(-)". Regarding clavicle abnormalities, use the full description "hypoplasia" and "agenesis" instead of H and A.
Now we have used the words “absent”, “hypoplasia” and “agenesis” instead of “(-), “H”, and “A” in table 1
- Where the authors describe the results of mutation analysis with prediction software, please avoid verbs such as "reveal" (lines 124, 226) in order to make it clear to the readers that in fact it's a prediction and there is no formal demonstration.
The word "reveal" was changed to "showed"
- please verify that the protein name is not in italics. All protein names were revised to avoid italics.
- please rephrase the legend to Fig. 2 and to Fig. 3:
Now the legends of figure 2 and 3 (now they are figure 4 and 5) were reformulated
"(A) The wild-type (Wt) p.Lys218 protein loses the nuclear localization site, proline-serine-threonine-rich domain, nuclear matrix targeting signal, and the conserved repression motif (white) in the p.Lys218Sfs*17 mutated (Mut) protein." Instead we suggest "The p.Lys218Sfs*17 mutation causes loss of the nuclear localization site, proline-serine-threonine-rich domain, nuclear matrix targeting signal,..."
Now we change the paragraph "(A) The wild-type (Wt) p.Lys218 protein loses the nuclear localization site, proline-serine-threonine-rich domain, nuclear matrix targeting signal, and the conserved repression motif (white) in the p.Lys218Sfs*17 mutated (Mut) protein” for the paragraph “(A) The p.Lys218Serfs*17 mutation (below) causes loss of the nuclear localization site, proline-serine-threonine-rich domain, nuclear matrix targeting signal present in the wild-type (Wt) p.Lys218 protein (above).”
Similarly, for "(B) Wild-type Ala180 protein adopts a folded conformation in the (C) p.Gln180 mutated protein. The wildtype p.Ala180 (blue) without bridges of hydrogens, in the mutated p.Gln180 (blue) gain a bridge of hydrogen bonds with p.Gln178 (yellow)." we think it's more appropriate "the p.Gln180 mutation causes the acquisition of a folding conformation owing to the possibility of hydrogen bonds between p.Gln180 and p.Gln178". Accordingly, modify also the legend to Figure 3.
The legends of figure 2 and 3 (now 4 and 5) have been modified for clarity
Reviewer 3 Report
Comments and Suggestions for Authors
This paper discusses newly observed genetic mutations in the RUNX2 gene in humans and the resulting clinical symptoms of Cleidocranial Dysplasia (CCD), as well as the molecular structural background. Thus, it is an interesting paper for understanding the mechanism of CCD. However, several points in the below should be addressed by the authors.
Major Revisions:
1. To enhance reader understanding, it is requested to include clear illustrations of the various domains of the RUNX2 gene product, in addition to textual descriptions in the current version of the manuscript. For this purpose, please refer to Figure 1 included in the following paper:
Thaweesapphithak, S., Theerapanon, T., Rattanapornsompong, K. et al. Functional consequences of C-terminal mutations in RUNX2. Sci Rep 13, 12202 (2023). https://doi.org/10.1038/s41598-023-39293-1
2. Descriptive explanations in Table 1 are insufficient as evidence for clinically observed findings from the novel mutations. Please present specific evidence such as X-ray images of dental anomalies within ethical boundaries.
Minor Revisions:
1. The title of the paper is somewhat abstract. Please reconsider for a more specific expression.
2. The English expressions used in the paper seem immature. Please undergo a native check.
3. Considering the context, perhaps the abbreviation list in the manuscript might not be necessary in this case. Please check this aspect once more.
Author Response
Thank you very much for taking the time to review this manuscript. Please find the detailed responses below and the corresponding revisions/corrections highlighted/in track changes in the re-submitted files.
Reviewer 3
This paper discusses newly observed genetic mutations in the RUNX2 gene in humans and the resulting clinical symptoms of Cleidocranial Dysplasia (CCD), as well as the molecular structural background. Thus, it is an interesting paper for understanding the mechanism of CCD. However, several points in the below should be addressed by the authors.
Major Revisions:
- To enhance reader understanding, it is requested to include clear illustrations of the various domains of the RUNX2 gene product, in addition to textual descriptions in the current version of the manuscript. For this purpose, please refer to Figure 1 included in the following paper:
Thaweesapphithak, S., Theerapanon, T., Rattanapornsompong, K. et al. Functional consequences of C-terminal mutations in RUNX2. Sci Rep 13, 12202 (2023). https://doi.org/10.1038/s41598-023-39293-1
Now we have added a schematic diagram (Figure 3) with the legend: “Figure 3. Position of novel genetic variants in the exons 3 and 4 (red boxes) of the RUNX2 gene and the position of amino acid changes in the domains RHD of the RUNX2 protein. The lower image shows the secondary structure of the protein with its domains QA (purple), RHD (blue), NLS (green), and NMTS (yellow).”
We also rely on the recommended reference by Thaweesapphithak, S et al. Scientific Report 13, 12202 (2023).
- Descriptive explanations in Table 1 are insufficient as evidence for clinically observed findings from the novel mutations. Please present specific evidence such as X-ray images of dental anomalies within ethical boundaries.
We have now added specific X-ray and imaging evidence of anomalies observed in case 1.
Minor Revisions:
- The title of the paper is somewhat abstract. Please reconsider for a more specific expression
We appreciate your suggestion but based on the observations of the other reviewers we decided to leave it the same.
- The English expressions used in the paper seem immature. Please undergo a native check.
Now the writing has been reviewed with a native proofreader through the “Scribendi” Editing and Proofreading service.
Considering the context, perhaps the abbreviation list in the manuscript might not be necessary in this case. Please check this aspect once more.
We have checked the abbreviations
Reviewer 4 Report
Comments and Suggestions for Authors
In this study Lopez et al. reported four variants in the RUNX2 gene associated with cleidocranial dysplasia. The authors characterized three of the four variants because these have not been yet described.
Some points need to be clarified.
Major points
Some pictures of the patients should be provided.
The new variants should be classified using accepted criteria such as those of ACMG.
For the classification of the variant of family 2 (Ala180Gln), genetic analysis of the parents of the proband is needed, to define if the variant occurred “de novo” or not. If the Ala180Gln variant is not “de novo”, it should be classified as VUS.
Clinical data of the relatives of the probands carrying the variants (family 1 and 4) should be reported with more details. In this version of the paper, these clinical data are just briefly described in the discussion.
The phenotypes occurred in subjects carrying the frame-shift variant should be compared with other patients carrying a frame-shift variant in RUNX2 (PMID: 35235174, PMID: 33538445).
Minor points
In ClinVar is present the variant Ala180Glu classified as pathogenic with one star. This could help the classification of the variant Ala180Gln.
In line 106 seems that mutation taster is a database of frequency. In addition, Exac no longer exist, and the data are now available in gnomAD. I checked the frequency of the new variants in gnomAD and as expected the variant are absent.
In line 192 the authors should assess more cautiously the effect of the frame-shift variant. “the effect produced by these mutations is of the negative dominance type”. Please write “could be” instead of “is”
The discussion starting from line 200 should be better written. It seems that the authors are talking about their own data.
Author Response
Thank you very much for taking the time to review this manuscript. Please find the detailed responses below and the corresponding revisions/corrections highlighted/in track changes in the re-submitted files.
Reviewer 4
In this study Lopez et al. reported four variants in the RUNX2 gene associated with cleidocranial dysplasia. The authors characterized three of the four variants because these have not been yet described.
Some points need to be clarified.
Major points
Some pictures of the patients should be provided.
In the figure 1, we have now added specific X-ray and imaging evidence of anomalies observed in our proband 1.
The new variants should be classified using accepted criteria such as those of ACMG. For the classification of the variant of family 2 (Ala180Gln), genetic analysis of the parents of the proband is needed, to define if the variant occurred “de novo” or not. If the Ala180Gln variant is not “de novo”, it should be classified as VUS.
The parents did not present clinical data of CCD, nor the new variant. In the results section we added the paragraph “The parents of probands 2 and 3 did not show the new variants.”. Furthermore, healthy controls did not present the variant, so we considered this variant to be de novo pathogenic.
Clinical data of the relatives of the probands carrying the variants (family 1 and 4) should be reported with more details. In this version of the paper, these clinical data are just briefly described in the discussion.
In the patient’s section we add a text summarizing the clinical data of the relatives of the probands carrying the variants (family 1 to 4): “The proband 1 (Figure 1A–F) and his mother showed a similar phenotype except that mother was absent of cataract, retinal damage, or facial changes. Data on the aunt were not obtained; it was only known that she had a diagnosis of CCD. With respect to proband 4, her brother and mother had similar characteristics without any headache, brachydactyly of fourth phalanx of the left foot, or facial anomalies. The parents of probands 2 and 3 showed no clinical abnormalities.”
The phenotypes occurred in subjects carrying the frame-shift variant should be compared with other patients carrying a frame-shift variant in RUNX2 (PMID: 35235174, PMID: 33538445).
Now we have made a further comparison with the recommended literature and added the text in the discussion section “A previous report of frameshift deletion was observed in a 15-year-old boy who presented a c.1550delT (p.Trp518Glyfs*60) mutation [25]. Another case of a 10-year-old boy with his mother showed a frameshift deletion 1554delG (p.Trp518Cysfs*61) [26], and the authors in both studies concluded that mutation affected the transactivation activity in the C-terminal zone of RUNX2. All patients in these studies presented typical CCD features.
The authors Gong L et al, 2022 and Zhao W, et al. 2021 were added to the text and the list of references with the numbers 26 and 27.
Minor points
In ClinVar is present the variant Ala180Glu classified as pathogenic with one star. This could help the classification of the variant Ala180Gln.
Indeed, in clinvar it is reported as a germline pathogenic variant without somatic clinical impact and it has not been reported in a previous article, so we consider it a new pathogenic variant.
In line 106 seems that mutation taster is a database of frequency. In addition, Exac no longer exist, and the data are now available in gnomAD. I checked the frequency of the new variants in gnomAD and as expected the variant are absent.
Indeed, we have checked and now we have changed “Exac” for “gnomAD”
In line 192 the authors should assess more cautiously the effect of the frame-shift variant. “the effect produced by these mutations is of the negative dominance type”. Please write “could be” instead of “is”
Now we have written the word "could be" instead of "is"
The discussion starting from line 200 should be better written. It seems that the authors are talking about their own data.
Now we have changed some words from line 200 (now line 269) to better spell the discussion and not be considered our data.
Answer to editor
A review of the references was carried out and it was adapted to the format of the journal.
By agreement of all the authors, Belen Ibarra Castrejón is added to the list of authors.
Round 2
Reviewer 1 Report
Comments and Suggestions for Authors
Comments on the Quality of English LanguageAuthor Response
Prof. Dr. Jukka Finne
Prof. Dr. Andrés Moya
Editors-in-chief
Biology
We were pleased to receive your letter concerning our manuscript entitled "New genetic variants of RUNX2 in Mexican families causes cleidocranial dysplasia" registered with the number biology-2868585 submitted to Biology. We now enclose a revised version R2 of this manuscript, which incorporates the excellent suggestions of the reviewer 1. Changes to the text are shown as underlined and highlighted in bold
Reviewer’s comments
The manuscript has been improved by the revision, although the reviewer regrets that the
authors have omitted functional assays and that there is a lack of attention to the content of
the writing.
- Figure 3 and figure legends: Although the authors have included a schematic diagram
illustrating the structure of RUNX2, the reviewer could not find "Runt", which is mentioned
in "domain location" in Table 1. Is "RHD" in Figure 3 the same as "Runt" in Table 1, or part
of it?
Now in table 1 the word “runt” was changed to “RHD” and to be homologated with RHD in figure 3, RDH has its meaning broken down at the end of table 1.
- Figure 3 Legend: Correct "the domains RHD". The authors mentioned that D of RHD is a
domain.
Now we change the phrase “the domains RHD” to the phrase “the RHD domain”
- Figure 3: What is CRM? Please define the abbreviations used in Figure 3 in the figure
legends. What is '1' in the left purple triangle?
In the introduction section we add the phrases “the first activation domain (1) [3]”; “and the conserved repressive motif (CRM), consisting of the amino acids valine-tryptophan-arginine-proline-tyrosine (VWRPY) [5].”
In the figure legend of figure 3 the abbreviations for QA: “Glutamine/alanine domain” (purple), RHD: “Runt homologous domain” (blue), NLS: “Nuclear localization signal region” (green), NMTS: “Nuclear matrix-targeting sequence domain” (yellow), “1: the first activation domain (purple triangle), CRM: the conserved repressive motif (orange rectangle).” were added.
- Figure 3: It is difficult to see the black characters in the blue box. The authors should
change one of the colors.
Now the background of figure 3 has been changed to white for better visualization
- Figure 4A: Please enter the domains in the same way as shown in the lower part of
Figure 3. In figure 4A the domains are indicated in the same way as in the lower part of figure 3
In the legend of Figure 4A “(A) The p.Lys218Serfs*17 mutation (below) causes loss of the nuclear localization site, proline-serine-threonine-rich domain, nuclear matrix targeting signal present in the wild-type (Wt) p.Lys218 protein (above)” was modified by legend “(A) The p.Lys218Serfs*17 mutation (below) causes modification of RHD and NLS with loss of PST, NMTS (inside the ellipses), and CRM present in the wild-type (Wt) p.Lys218 protein (above)”
- Although the authors mentioned in the rebuttal letter that "We have now corrected the
variant name "p.Lys218Sfs*17" to "p.Lys218Serfs*17", there are still inappropriate
sequence variants "p.Lys218Sfs*17" and "p.Lys218Sfer*17" in the revised manuscript.
Thanking you in advance for your valuable comments, we continue to correct the errors with the correct “p.Lys218Serfs*17”
- The description of sequence variants, is now also inappropriate
(Hum Mutat. 2016 Jun;37(6):564-9. doi: 10.1002/humu.22981). Please correct.
“p.Gln67X” or “p.Gly462X” were changed to “p.Gln67*” and “p.Gly462*” respectively.
- Are “Q/A” and “QA” the same?
Now we have left only “QA”, eliminating the diagonal
- Table 2: There is still an extra space in the primer sequence.
Corrected
- Line 275: Change "1554delG" to "c. 1554delG".
Now we change "1554delG" to "c.1554delG”
- Line 309: Please add the description CBF before the description of the impaired
transactivation activity.
Now we have added the description “Core Binding Factor β”
- CBF is not the only protein that interacts with RUNX2 via the Runt domain; however, the
authors only focused on this molecule in the Discussion. Is there a reason for this? If not,
please add the information about other transcription factors and discuss with references.
Now we have added in the discussion a paragraph related to the interaction of proteins or transcription factors with the runt domain of RUNX2 and the importance of CBFβ: “[23]; the union of RUNX2 with its partner; the cofactor CBFβ is very important since it allows to improve its DNA binding affinity by promoting structural changes that reveal the RUNX2-DNA interaction surface [28], it also protects and stabilizes against proteolytic degradation [29]. Some factors such as Msh homeobox 2 (Msx2), twist or promyelocy-tic leukemia zinc finger protein (PLZF) and osterix (Osx) that participate in the induction of RUNX2 expression [30,31], can be affected by their bad interaction with the runt domain, likewise RHD damage downregulates the transactivation activity, decreasing the expression of bone matrix genes, including collagen I, osteocalcin, bone sialoprotein, and osteopontin [3,5,23].”
- In most instances, we prefer the use of ‘person’ or ‘individual’ rather than ‘patient’. Please
revise accordingly.
The term “patient” was changed by terms such as “proband”, “person” or “individual”.
Note: Regarding the sending of the English edition, we comment that it has already been sent to edition with a Native English-speaking editor at "Scribendi Customer Service" with order number # 992160, access code tw4YyGqhPXMw.
Reviewer 4 Report
Comments and Suggestions for Authors
The quality of the manuscript is significantly improved
Author Response
Prof. Dr. Jukka Finne
Prof. Dr. Andrés Moya
Editors-in-chief
Biology
We are grateful for your comment